# Effect of ultrasonography surveillance in patients with liver cancer: a population-based longitudinal study

Jui-Kun Chiang,[1] Lin Chih-Wen,[2,3] Yee-Hsin Kao[4]

[1]Department of Family Medicine, Dalin Tzu Chi Hospital, Buddhist Tzu Chi Medical Foundation, Dalin, Chiayi, Taiwan
[2]Department of Medical Imaging, Dalin Tzu Chi Hospital, Buddhist Tzu Chi Medical Foundation, Dalin, Chiayi, Taiwan
[3]School of Medicine, Tzu Chi University, Hualien, Taiwan
[4]Department of Family Medicine, Tainan Municipal Hospital, Tainan, Taiwan

**Correspondence to**
Dr Yee-Hsin Kao;
m2200767@gmail.com

## ABSTRACT

**Objective** Liver cancer is a growing global public health problem. Ultrasonography is an imaging tool widely used for the early diagnosis of liver cancer. However, the effect of ultrasonography surveillance (US) on the survival of patients with liver cancer is unknown. Therefore, this study examined the association between survival and US frequency during the 2 years preceding patients' liver cancer diagnosis.

**Methods** This population-based longitudinal study was conducted in Taiwan, a region with high liver cancer incidence, by using the National Health Insurance Research Database. We compared survival between patients who received US three times or more (≥3 group) and less than three times (<3 group) during the 2 years preceding their liver cancer diagnosis, and identified the predictors for the ≥3 group.

**Results** This study enrolled 4621 patients with liver cancer who had died between 1997 and 2010. The median survival rate was higher in the ≥3 group (1.42 years) than in the <3 group (0.51 years). Five-year survival probability was also significantly higher in the ≥3 group (14.4%) than in the <3 group (7.7%). The multivariate logistic regression results showed that the three most common positive predictors for receiving three or more US sessions were indications of viral hepatitis, gallbladder diseases and kidney–urinary–bladder diseases; the most common negative predictors for receiving three or more US sessions were male sex and indications of abdominal pain.

**Conclusion** Patients with liver cancer who received US three times or more during the 2 years preceding their liver cancer diagnosis exhibited a higher 5-year survival probability.

## INTRODUCTION

Liver cancer is a growing global public health problem and is the fifth and ninth most common cancer in men and women worldwide, respectively. Because of its poor outcome, it is also the second most common cause of cancer-related mortality, accounting for nearly 746 000 deaths worldwide in 2012.[1] Eastern and Southeastern Asia, including Taiwan, are regions with a particularly high incidence of liver cancer. In Taiwan, liver cancer has been the second leading cause of cancer-related mortality since 2004, and it accounted for 8116 (18.6%) of the 43 665

### Strengths and limitations of this study

► This is a nationwide, population-based study that was conducted in Taiwan, a region that has a high incidence of liver cancer.
► We determined that patients' 5-year survival probability was significantly higher among those who had received three or more sessions of ultrasonography surveillance (US) (14.4%) than among those who received less than three sessions of US (7.7%).
► The survival results associated with patients receiving three or more sessions of US 2 years prior to their liver cancer diagnosis may be applicable to the general population in Taiwan because of the high incidence of liver cancer.
► No information on cancer stage was available in the dataset, and some crucial potentially confounding variables were also not available for performing statistical adjustment.
► The cause of death was not recorded in the insurance claims data, and therefore we cannot establish associations between liver cancer and mortality.

cancer deaths in 2012.[2] The known risk factors for liver cancer are viral infection (hepatitis B virus (HBV) and hepatitis C virus (HCV) infections), alcohol consumption and alpha toxins, diabetes, and metabolic (eg, non-alcoholic fatty liver disease (NAFLD) and hereditary haemochromatosis) and immune-related (eg, primary biliary cirrhosis and autoimmune hepatitis) diseases.[3]

One previous study reported that chronic HBV and HCV infections are the primary risk factors for liver cancer.[4] To improve the effective control and treatment of viral hepatitis, two programmes have been implemented in Taiwan. The first is the mass vaccination programme targeted against HBV, which has considerably reduced the HBV carrier rate among children and adolescents and consequently reduced the incidence of childhood liver cancer in Taiwan.[5–8] The second programme is a pilot programme initiated

in October 2003 by the Taiwanese Center for Disease Control, entitled 'Strengthening of treatment for chronic hepatitis B and C under the National Health Insurance'. This programme has attempted to ensure that patients with chronic hepatitis B and C are registered, so that they can benefit from appropriate diagnosis, monitoring and treatment.[9] Accordingly, the incidence of liver cancer significantly decreased by an average annual percentage of 1.1% from 2002 to 2012; during this same period, the 5-year relative survival rates for liver cancer were 52.0% for stage I, 2.9% for stage IV and 28.9% for all stages.[10]

A diagnosis of hepatocellular cancer is based on a combination of radiological, serological and histopathological criteria. Some studies have reported that alpha-fetoprotein determination lacks adequate sensitivity and specificity for effective surveillance[11 12]; therefore, surveillance must be based on ultrasonography, which has a recommended screening interval of 6 months.[13 14] A previous study reported that ultrasonography has a sensitivity of approximately 63% and a high specificity of approximately 85%–90% for the detection of early-stage liver cancer.[12] Moreover, ultrasonography is a non-invasive and safe method, is well accepted by patients and is relatively inexpensive.

Research has indicated that most patients with hepatocellular carcinoma have a low rate of survival. However, patients with early-stage hepatocellular carcinoma who receive potentially curative therapy exhibit considerably improved survival (5-year survival=40%–70%).[15] One systemic review provided very-low-strength evidence about the effects of hepatocellular carcinoma screening on mortality in patients with chronic hepatitis.[16] Nevertheless, regular surveillance with abdominal ultrasonography might facilitate the early detection of liver cancer, thus improving the survival of at-risk patients. This study therefore examined the effect of ultrasonography surveillance (US) on survival. Specifically, the association between patient survival and the frequency of US during the 2 years preceding liver cancer diagnosis was determined, and the predictors of receiving more US sessions were identified.

## METHODS

### Data source and patient identification

For this nationwide population-based study, we analysed the data from the National Health Insurance Research Database (NHIRD) of Taiwan. The National Health Insurance (NHI) programme, established in 1995, is a single-payer health insurance system that covers 99.9% of all residents in Taiwan. Moreover, 97% of medical providers nationwide are affiliated with the programme.[17] The Longitudinal Health Insurance Database (LHID) is a nationwide representative database containing all of the original claims data for the period of 1996–2011 for 1 million NHI beneficiaries randomly sampled from the 23.22 million NHI enrollees. The availability of this population-based database has stimulated

and facilitated academic research in various scientific disciplines, particularly in the field of health research.[18] In addition, the accuracy of diagnoses registered in the NHIRD has been validated using five-digit International Classification of Diseases, Ninth Revision, Clinical Modification (ICD-9-CM) codes, including those for diabetes and cancer, which confirms the reliability of the data source.[19] In Taiwan, all patients with cancer are designated as having a catastrophic illness. For this study, we identified patients with liver cancer and catastrophic illness designations in the LHID and followed up with these patients until December 2011. ICD-9-CM and A codes were used to define liver cancer (namely, 155, 155.0, 155.1 and A095).

### Study sample

This study included patients aged ≥18 years who had received a first-time diagnosis of liver cancer between 1 January 1997 and 31 December 2010. The NHIRD was used to identify the patients, and their diagnoses of cancer were confirmed using the Registry of Catastrophic Illness. Patients were excluded if they had made no insurance claims during their final year of life, were still alive during the 6 months before the end of the dataset or had missing data. A total of 4621 patients with liver cancer were enrolled in this study (figure 1).

### Frequency of abdominal ultrasonography surveillance during the 2 years preceding liver cancer diagnosis

We classified the study population into two groups based on their frequency of receiving US during the 2 years preceding their liver cancer diagnosis: the <3 group (patients who had received US less than three times; n=3149) and the ≥3 group (patients who had received US three times or more; n=1472).

We classified the diseases that are indicators for ultrasonography into seven groups using their respective ICD-9-CM diagnosis codes. The seven disease groups are as follows: (1) liver-related diseases, including chronic hepatitis, cirrhosis and other liver disorders (ICD-9 codes: 571, 571.x, 572, 572.0, 573 and 573.x); (2) viral hepatitis infections (ICD-9 codes for HBV: V0261, 070.20–070.23 and 070.30–070.33; for HCV: V0262, 070.41, 070.44, 070.51 and 070.54); (3) gallbladder and biliary tract-related diseases (ICD-9 codes: 574–576); (4) abdominal pain conditions (ICD-9 codes: 789.0x, 789.4x, 789.6x and 789.9); (5) kidney–urinary–bladder (KUB) diseases (ICD-9 codes: 591, 592, 592.0, 592.1, 592.9, 593 and 593.2); (6) pancreatic diseases (ICD-9 code: 577.x); and (7) other conditions, such as hepatomegaly, splenomegaly, abdominal or pelvic mass, and ascites (ICD-9 codes: 7891, 7892, 7893x and 7895).

### Definition of variables

Charlson comorbidity index (CCI) scores were calculated by examining patients' ICD-9-CM diagnoses and procedure codes recorded in the year prior to their liver cancer diagnosis, according to the Deyo method. After referring

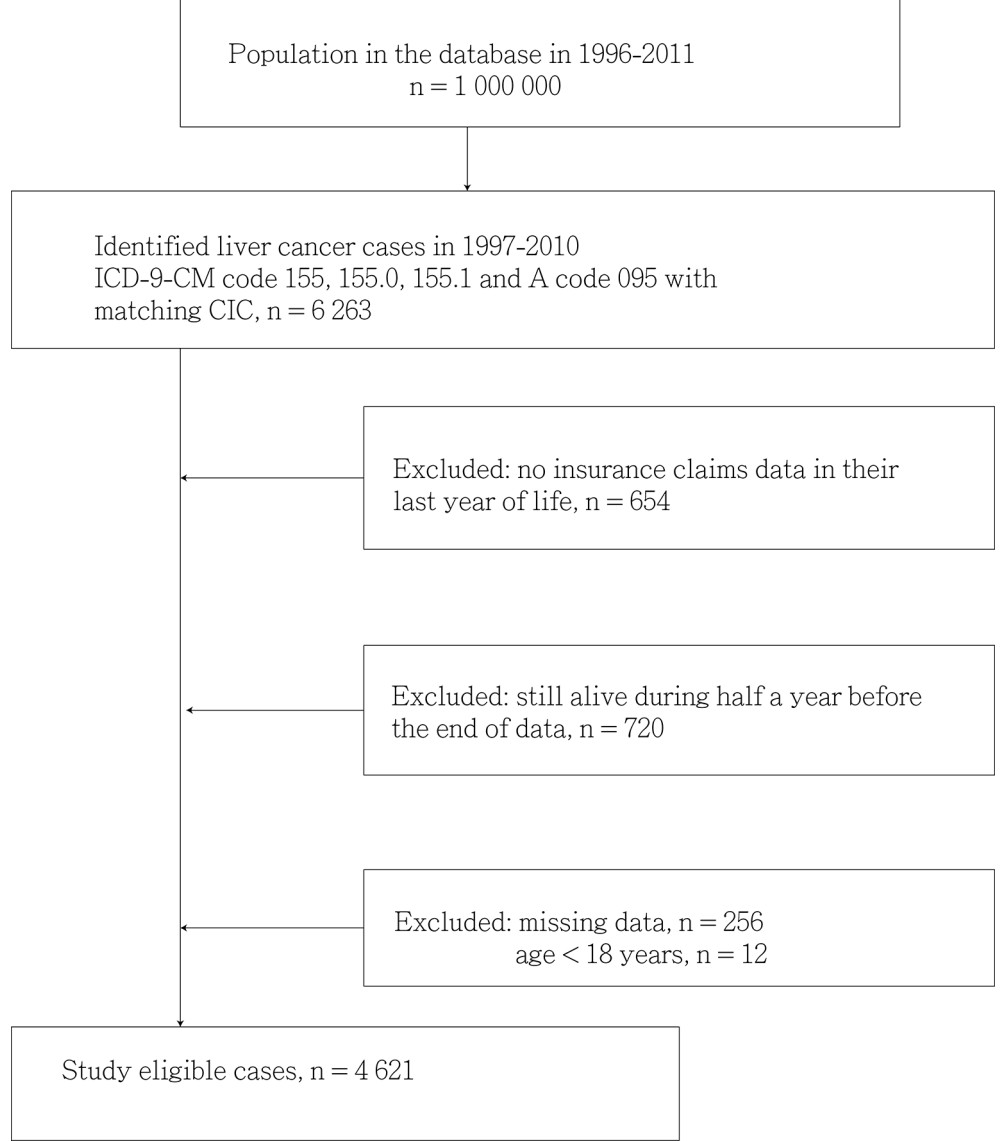

**Figure 1**  Study flowchart. ICD-9-CM, International Classification of Diseases, Ninth Revision, Clinical Modification.

to Klabundle *et al*, we calculated CCI scores for both inpatient and outpatient claims.[20–22]

The study protocol was reviewed and approved by the Research Ethics Committee of Dalin Tzu Chi Hospital, Buddhist Tzu Chi Medical Foundation, Taiwan (Approval No. B10304018). Because the NHIRD contains only deidentified secondary data, the review board waived the requirement for informed consent.

## Statistical analyses

In this paper, continuous variables are expressed in mean±D, and categorical variables are presented in frequency and percentage. For the univariate analysis, a two-sample t-test, Wilcoxon rank-sum test, $\chi^2$ test and Fisher's exact test were performed to examine differences in the distribution of the continuous and categorical variables between the two groups. The survival duration (years) was defined as the duration from the day of diagnosis to the day of death. In addition, generalised additive models (GAMs) were fitted to detect the potential non-linear effects of continuous covariates and to identify the appropriate cut-off points for discretising continuous covariates (if necessary) during the stepwise variable selection. Computationally, the VGAM function (with the default values of smoothing parameters) of the VGAM package was used to fit GAMs for the binary outcome in R.

Cox proportional hazard regression models were used to estimate the event-specific HRs and 95% CIs of all the variables listed in tables 1 and 2. Subsequently, a multivariate analysis was conducted by fitting the multiple logistic regression models (through a stepwise variable selection procedure) to identify the crucial predictors of receiving US three times or more during the 2 years preceding a liver cancer diagnosis. The goodness-of-fit (GOF) of the final logistic regression model was assessed by estimating the area under the curve (AUC; also called the c statistic), where $0 \le c \le 1$, and by conducting the Hosmer-Lemeshow GOF test. In practice, c ≥0.7 is an acceptable level of

**Table 1** Comparison of demographic characteristics between the <3 and ≥3 groups

| Variable | <3 group, n (%) | ≥3 group, n (%) | p Value |
|---|---|---|---|
| Total no (%) | 3149 (68.1) | 1472 (31.9) | |
| Age (years) | 63.04±13.92 | 63.77±11.76 | 0.474 |
| Gender | | | <0.001 |
| Female | 807 (25.6) | 527 (35.8) | |
| Male | 2342 (74.4) | 945 (64.2) | |
| Survival* | 0.51 (0.17, 1.81) | 1.42 (0.48, 3.34) | <0.001 |
| HBV | 1119 (35.5) | 619 (42.1) | <0.001 |
| HCV | 660 (21.0) | 620 (42.1) | <0.001 |
| Diabetes | 535 (17.0) | 361 (24.5) | <0.001 |
| CKD | 276 (8.8) | 232 (15.8) | <0.001 |
| CVA | 535 (17.0) | 297 (20.2) | 0.001 |
| Hypertension | 1173 (37.2) | 691 (46.9) | <0.001 |
| Cirrhosis | 2048 (65.0) | 1183 (80.4) | <0.001 |
| Employment (yes) | 1873 (59.5) | 891 (60.5) | 0.499 |
| Northern area in Taiwan | 1063 (33.8) | 467 (31.7) | 0.178 |
| Central area in Taiwan | 888 (28.2) | 461 (31.3) | 0.031 |
| Southern area in Taiwan | 1032 (32.8) | 480 (32.6) | 0.920 |
| Eastern area in Taiwan | 153 (4.9) | 57 (3.9) | 0.150 |

*Survival: median (first quartile, third quartile).
CCI, Charlson comorbidity index; CKD, chronic kidney disease; CVA, cerebral vascular accident; HBV, hepatitis B virus; HCV, hepatitis C virus.

discrimination power for a fitted logistic regression model and p >0.05 in the Hosmer-Lemeshow test indicates a good fit for the logistic regression model. Moreover, a variance inflation factor of ≥10 for continuous covariates or ≥2.5 for categorical covariates indicates that multicollinearity occurs among some of the covariates in a fitted logistic regression model. All statistical analyses were performed

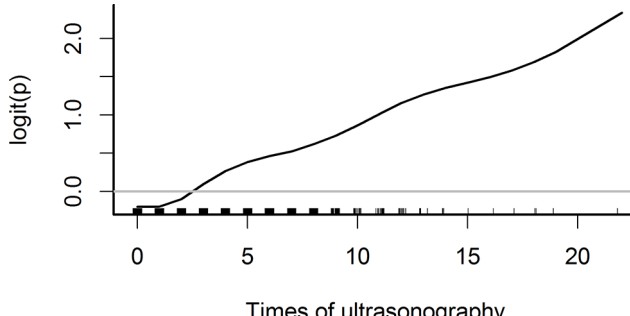

**Figure 2** Smoothing curve of the frequency of ultrasonography during the 2 years preceding liver cancer diagnosis against 5-year survival probability after adjustment (x-axis: frequency of ultrasonography during the 2 years preceding liver cancer diagnosis; y-axis: probability of 5-year survival).

using the R V.3.0.2 software (R Foundation for Statistical Computing). Statistical significance was set at p ≤0.05.

## RESULTS

We enrolled 4621 adult patients (3287 men and 1334 women; ratio=2.46:1) with liver cancer who died during 1997–2010. Figure 1 depicts the study design. Specifically, we explored the association between the frequency of US during the 2 years preceding a patient's liver cancer diagnosis and that patient's 5-year survival probability. The optimal cut-off point for receiving US to ensure a higher 5-year survival probability was set at three sessions (figure 2).

Table 1 summarises the demographic characteristics of the study patients and indicates that those in the ≥3 group were more likely to be women (p<0.001) and were more likely to have the comorbidities of viral hepatitis (HBV or HCV infection) (p<0.001), cirrhosis (p<0.001), diabetes (p<0.001), hypertension (p<0.001) or stroke (p=0.001). Notably, the median survival of the ≥3 group (1.42 years) was higher than that of the <3 group (0.51 years) (p<0.001). The indicators for abdominal US used in this study are summarised in table 2. The most frequent indicators were viral hepatitis (2580, 55.8%), gallbladder diseases (1690, 36.6%) and abdominal pain (1387,

**Table 2** Ultrasonography surveillance indicators in the <3 and ≥3 groups

| Variable | Total no (%) | <3 group, n (%) | ≥3 group, n (%) | p Value |
|---|---|---|---|---|
| Viral hepatitis (HBV, HCV) | 2580 (55.8) | 1556 (49.4) | 1024 (69.6) | <0.001 |
| Chronic hepatitis (except HBV, HCV) | 1560 (33.8) | 1179 (37.4) | 381 (25.9) | <0.001 |
| Gallbladder diseases | 1690 (36.6) | 989 (31.4) | 701 (47.6) | <0.001 |
| Abdominal pain | 1387 (30.0) | 921 (29.2) | 466 (31.7) | 0.098 |
| KUB diseases | 928 (20.1) | 524 (16.6) | 404 (27.4) | <0.001 |
| Pancreas diseases | 271 (5.9) | 133 (4.2) | 138 (9.4) | <0.001 |
| Others* | 1780 (38.5) | 1085 (34.5) | 695 (47.2) | <0.001 |

*Hepatomegaly, splenomegaly, abdominal or pelvic mass, and ascites.
HBV, hepatitis B virus; HCV, hepatitis C virus; KUB, kidney–urinary–bladder.

**Table 3** Comparison of treatment received by the <3 and ≥3 groups following ultrasonography surveillance

| Variable | <3 group, n (%) | ≥3 group, n (%) | p Value |
|---|---|---|---|
| PEI | 200 (6.4) | 244 (16.6) | <0.001 |
| Hepatectomy | 343 (10.9) | 221 (15.0) | <0.001 |
| RFA | 71 (2.3) | 101 (6.9) | <0.001 |
| Liver transplantation | 2 (0.1) | 6 (0.4) | 0.015 |
| TACE | 584 (18.5) | 453 (30.8) | <0.001 |
| Chemotherapy | 1169 (37.1) | 677 (46.0) | <0.001 |
| Radiotherapy | 647 (20.5) | 334 (22.7) | 0.097 |
| Five-year survival probability | 7.7% | 14.4% | <0.001 |

PEI, percutaneous ethanol injection; RFA, radiofrequency ablation; TACE, transcatheter arterial chemoembolisation.

30.0%). In this study, all indicators, except for abdominal pain (p=0.098), were significantly higher in the ≥3 group than in the <3 group.

Compared with the <3 group, significantly higher proportions of the ≥3 group received all possible treatments for liver cancer, except for radiotherapy (p=0.097); the ≥3 group also had significantly higher 5-year survival probability than did the <3 group (14.4% vs 7.7%, respectively; p<0.001) (table 3). From the multivariate Cox proportional hazard regression model, the positive significant predictors of death were male sex (HR 1.14, 95% CI 1.07 to 1.22), older age (HR 1.02, 95% CI 1.01 to 1.03), diabetes (HR 1.13, 95% CI 1.04 to 1.22) and a high CCI score (HR 1.07, 95% CI 1.05 to 1.08). Conversely, the negative significant predictors of death were hypertension (HR 0.83, 95% CI 0.78 to 0.89), stroke (HR 0.78, 95% CI 0.72 to 0.84), chronic kidney diseases (CKDs) (HR 0.84, 95% CI 0.74 to 0.92), viral hepatitis (HBV or HCV infection) (HR 0.79, 95% CI 0.72 to 0.88) and receiving US three times or more during the 2 years preceding liver cancer diagnosis (HR 0.80, 95% CI 0.75 to 0.85).

The significant indicators for US among the patients in this study included KUB diseases (HR 0.92, 95% CI 0.86 to 0.995), gallbladder diseases (HR 0.90, 95% CI 0.85 to 0.96), chronic hepatitis (except for HBV or HCV infection) (HR 0.90, 95% CI 0.81 to 0.999) and pancreatic diseases (HR 0.80, 95% CI 0.71 to 0.91). Moreover, the primary complications of liver cancer were upper gastrointestinal bleeding (gastric ulcer or duodenal ulcer) (HR 0.92, 95% CI 0.86 to 0.98) and oesophageal variceal bleeding (HR 0.90, 95% CI 0.83 to 0.97), and the main treatments that the patients received for liver cancer were transcatheter arterial chemoembolisation (TACE) (HR 0.82, 95% CI 0.76 to 0.89), radiotherapy (HR 0.81, 95% CI 0.75 to 0.87), radiofrequency ablation (RFA) (HR 0.79, 95% CI 0.68 to 0.93), chemotherapy (HR 0.71, 95% CI 0.67 to 0.76), resection (HR 0.67, 95% CI 0.61 to 0.73) and percutaneous ethanol injection (PEI) (HR 0.64, 95% CI 0.57 to 0.70). Furthermore, the demographic

factors significantly linked with greater US frequency were employment (HR 0.89, 95% CI 0.83 to 0.95) and living in Northern Taiwan (HR 0.91, 95% CI 0.85 to 0.97) (table 4).

Survival probability was significantly higher in the ≥3 group than in the <3 group (p<0.001; figure 3). According to the multivariate logistic regression model, the significant positive predictors for survival in the ≥3 group were viral hepatitis (HBV or HCV infection) (OR 3.66, 95% CI 2.76 to 4.85), gallbladder diseases (OR 1.82, 95% CI 1.59 to 2.09), KUB diseases (OR 1.80, 95% CI 1.53 to 2.11), pancreatic diseases (OR 1.79, 95% CI 1.37 to 2.34), chronic hepatitis (except for viral hepatitis; OR 1.75, 95% CI 1.31 to 2.34), CKDs (OR 1.60, 95% CI 1.30 to 1.96), diabetes (OR 1.56, 95% CI 1.31 to 1.86), US indications of other conditions (OR 1.47, 95% CI 1.29 to 1.69), hypertension (OR 1.17, 95% CI 1.01 to 1.34) and living in Central Taiwan (OR 1.16, 95% CI 1.01 to 1.34). Conversely, the significant negative predictors for survival in the ≥3 group were male sex (OR 0.60, 95% CI 0.52 to 0.69) and the US indication of abdominal pain (OR 0.81, 95% CI 0.70 to 0.93) (table 5).

The results of the Hosmer-Lemeshow test indicated a good fit (p=0.156) and that the AUC was acceptable (0.707, 95% CI 0.691 to 0.723) (figure 4).

## DISCUSSION

In this study, we found that patients with liver cancer who had received US three times or more during the 2 years preceding their liver cancer diagnosis exhibited high 5-year survival compared with those who had received US less than three times during that same period (14.4% vs 7.7%). We also determined that, compared with the <3 group, higher proportions of the ≥3 group received treatment for early-stage liver cancer (eg, PEI, hepatectomy, RFA or liver transplantation). As illustrated in figure 2, there was a linear dose–response relationship between US frequency and the logit of 5-year survival probability; a similar result was also reported in a previous study.[23] Further cost-effectiveness analyses are needed to justify recommending three or more sessions of US to people at a high risk of liver cancer diagnosis or to the general population in an area with a high incidence of liver cancer.

According to one previous trial, biannual US for patients with HBV infection reduces hepatocellular carcinoma mortality by 37%.[24] Another cohort study reported that patients with viral hepatitis who receive routine US exhibit higher 5-year survival than do those who did not receive routine US (31.84% vs 20.67%, respectively).[25] Other scholars have indicated that certain populations at a high risk for developing hepatocellular carcinoma benefit from more intensive US.[26–30] One trial study revealed that early-stage hepatocellular carcinoma was more likely to be detected in patients with viral hepatitis who received US at 4-month intervals than in those who received US at 12-month intervals; however, patients' overall survival was

**Table 4** Results of the multivariate Cox proportional hazard regression of predictors of death among patients with liver cancer

| Variable | HR | 95% CI | p Value |
|---|---|---|---|
| Male | 1.14 | 1.07 to 1.22 | <0.001 |
| Age (hepatocellular cancer diagnosis) per 5 years | 1.02 | 1.01 to 1.03 | 0.002 |
| Diabetes | 1.13 | 1.04 to 1.22 | 0.004 |
| Hypertension | 0.83 | 0.78 to 0.89 | <0.001 |
| CVA | 0.78 | 0.72 to 0.84 | <0.001 |
| CKD | 0.84 | 0.74 to 0.92 | <0.001 |
| Viral hepatitis (HBV, HCV) | 0.79 | 0.72 to 0.88 | <0.001 |
| CCI (per 1 score) | 1.07 | 1.05 to 1.08 | <0.001 |
| US indications | | | |
| Three times or more US | 0.80 | 0.75 to 0.85 | <0.001 |
| KUB diseases | 0.92 | 0.86 to 0.995 | 0.037 |
| Gallbladder diseases | 0.90 | 0.85 to 0.96 | 0.001 |
| Chronic liver diseases (except HBV, HCV) | 0.90 | 0.81 to 0.999 | 0.047 |
| Pancreas diseases | 0.80 | 0.71 to 0.91 | <0.001 |
| Complication of liver cancer | | | |
| UGI bleeding (GU, DU) | 0.92 | 0.86 to 0.98 | 0.009 |
| EV bleeding | 0.90 | 0.83 to 0.97 | 0.008 |
| Treatments | | | |
| TACE | 0.82 | 0.76 to 0.89 | <0.001 |
| Radiotherapy | 0.81 | 0.75 to 0.87 | <0.001 |
| RFA | 0.79 | 0.68 to 0.93 | 0.004 |
| Chemotherapy | 0.71 | 0.67 to 0.76 | <0.001 |
| hepatectomy | 0.67 | 0.61 to 0.73 | <0.001 |
| PEI | 0.64 | 0.57 to 0.70 | <0.001 |
| Background | | | |
| Employment | 0.89 | 0.83 to 0.95 | <0.001 |
| Northern area in Taiwan | 0.91 | 0.85 to 0.97 | 0.003 |

CCI, Charlson comorbidity index; CKD, chronic kidney disease; CVA, cerebral vascular accident; DU, duodenal ulcer; EV, esophageal varices; GU, gastric ulcer; HBV, hepatitis B virus; HCV, hepatitis C virus; KUB, kidney–urinary–bladder; PEI, percutaneous ethanol injection; RFA, radiofrequency ablation; TACE, transcatheter arterial chemoembolisation; UGI, upper gastrointestinal tract; US, ultrasonography surveillance.

not different at the 4-year follow-up.[31] In another systemic review, the effects of hepatocellular carcinoma screening on mortality in patients with chronic hepatitis were found to be uncertain.[16]

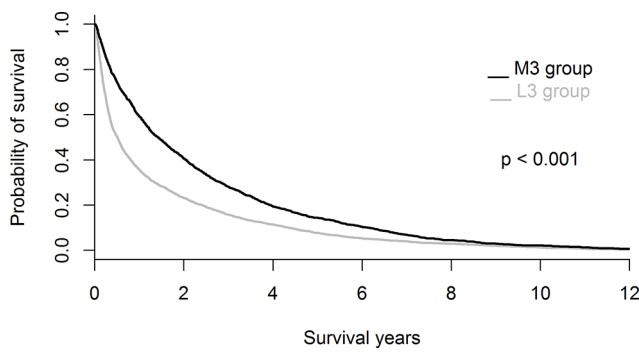

**Figure 3** Survival curves of the <3 group and ≥3 group.

In short, receiving US at a high frequency has no confirmed beneficial effect on the 5-year survival probability of patients with hepatocellular carcinoma. This finding may be because the early detection of liver cancer improves the rate of curative treatments. However, the prognosis of patients with liver cancer is not solely related to the tumour stage,[32] and regular follow-up for possible recurrence and treatment is vital. Although the survival of patients with liver cancer has improved overall in the past decade, physicians caring for at-risk patients should provide regular high-quality screening, including ultrasonography.[13]

Although advanced diagnostic techniques, such as dynamic multiphasic multidetector-row CT and MRI, are the standard diagnostic methods for the non-invasive diagnosis of liver cancer, ultrasonography still plays a crucial role in liver cancer surveillance.[33] In Taiwan,

**Table 5** Results of the multivariate logistic regression of the predictors for receiving ultrasonography surveillance three times or more during the 2 years preceding a liver cancer diagnosis

| Variables | OR | 95% CI | p Value |
|---|---|---|---|
| Male | 0.60 | 0.52 to 0.69 | <0.001 |
| Diabetes | 1.56 | 1.31 to 1.86 | <0.001 |
| Hypertension | 1.17 | 1.01 to 1.34 | 0.03 |
| CKD | 1.60 | 1.30 to 1.96 | <0.001 |
| US indications | | | |
| Viral hepatitis (HBV or HCV infection) | 3.66 | 2.76 to 4.85 | <0.001 |
| Gallbladder diseases | 1.82 | 1.59 to 2.09 | <0.001 |
| KUB diseases | 1.80 | 1.53 to 2.11 | <0.001 |
| Pancreas diseases | 1.79 | 1.37 to 2.34 | <0.001 |
| Chronic hepatitis except viral hepatitis | 1.75 | 1.31 to 2.34 | <0.001 |
| Other* | 1.47 | 1.29 to 1.69 | <0.001 |
| Abdominal pain | 0.81 | 0.70 to 0.93 | 0.004 |
| Central area in Taiwan | 1.16 | 1.01 to 1.34 | 0.04 |

*Includes hepatomegaly, splenomegaly, abdominal or pelvic mass, and ascites.
CKD, chronic kidney disease; HBV, hepatitis B virus; HCV, hepatitis C virus; KUB, kidney–urinary–bladder; US, ultrasonography surveillance.

the recommended intervals for US screening is 6 months for patients with viral hepatitis infection and 3 months for patients with cirrhosis. Ultrasonography can be performed in hospital and clinics, if indicated, and the cost of each ultrasonography screening (US$23.1) is covered by the NHI programme.

In the present study, we found that patients with chronic hepatitis (both HBV and HCV, and non-HBV and non-HCV infections), diabetes, CKDs, cerebral vascular accident (CVA), hypertension, gallbladder diseases, KUB diseases, pancreatic diseases and other US-indicating

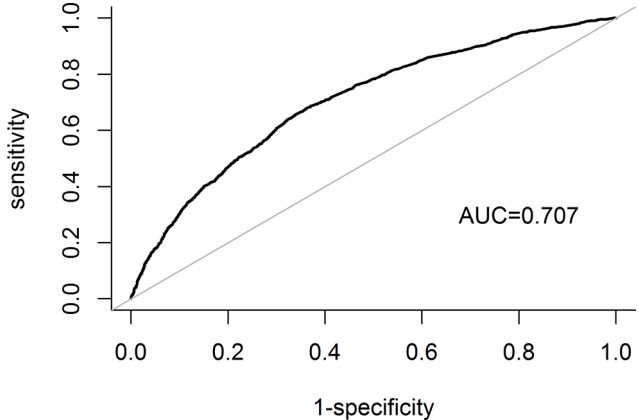

**Figure 4** Receiver operating characteristic curves of the predictors for receiving ultrasonography three or more times during the 2 years preceding liver cancer diagnosis. AUC, area under the curve.

conditions (ie, hepatomegaly, splenomegaly, abdominal or pelvic mass, and ascites) were more likely to receive US three times or more during the 2 years preceding their liver cancer diagnosis. One reason that patients with diabetes in this study were more likely to receive US three times or more might be because these patients are at a high risk of primary liver cancer.[34–36] Notably, the association between diabetes and liver cancer is significantly strengthened when the history of diabetes is longer than 10 years.[37 38] Some studies have also suggested that insulin might enhance the risk of hepatocellular cancer, whereas certain oral glucose-lowering medications and statins may decrease the risk of hepatocellular cancer.[39 40]

Similarly, patients with CKD in this study were more likely to receive US three times or more, most likely because of the close link between kidney diseases and cancer.[41] Indeed, cancer has become increasingly recognised as a complication and a major cause of morbidity and mortality in the CKD population.[42] Finally, the reason that patients with CVA, hypertension, gallbladder diseases, KUB diseases, pancreatic diseases or other US-indicating conditions in this study were more likely to receive US three times or more is probably because patients with more comorbidities and physician visits are more likely to undergo more laboratory tests and imaging studies, including US, if they experience any discomfort.

The chances of developing NAFLD are enhanced by type 2 diabetes, arterial hypertension, obesity, metabolic syndrome, mixed hyperlipidaemia and familial hypobeta-lipoproteinaemia, all of which are the chief metabolic modifiers of NAFLD risk.[43–45] In this study, NAFLD was included in the group of liver-related diseases alongside chronic hepatitis, cirrhosis and other liver disorders. However, the imprecise classification for NAFLD is one of the claims-data-related limitations of this study. Instead, we classified the diseases that are the indicators for US into seven groups; patients with these indicators easily received US due to its widespread availability in Taiwan. Another limitation of this study is that baseline data were not recorded in the claims data and thus could not be investigated. For example, the underlying causes of pancreatic disease include alcohol abuse, biliary disease, hyperlipidaemia and other risk factors, aetiologies or idiopathic diseases[46]; however, it is unknown whether patients in the present study who had pancreatic disease were also diagnosed with any of these baseline factors or diseases.

As noted earlier, higher proportions of the ≥3 group received treatment for early-stage liver cancer, such as PEI, hepatectomy, RFA or liver transplantation, than did the <3 group. Such treatments have similarly been reported as therapies for early-stage hepatocellular carcinoma in previous studies.[47–50] We also determined that higher proportions of the ≥3 group received TACE compared with the <3 group, although the therapeutic effects of TACE for early-stage hepatocellular carcinoma are unclear.[51]

One previous study reported that mass screening for liver cancer by using US in high-endemic regions,

including Taiwan, is cost-effective and recommended.[52] In this study, the significant positive predictors for survival in the ≥3 group were viral hepatitis, gallbladder diseases, KUB diseases, pancreatic diseases, chronic hepatitis (except viral infections), diabetes, US indications of other conditions, hypertension and living in Central Taiwan. By contrast, the significant negative predictors for survival in the ≥3 group were male sex and the US indication of abdominal pain. The reason that these factors are negative predictors might be because men are less likely to seek medical attention than are women,[53] and because after seeking medical attention for one abdominal pain event, patients do not return for subsequent follow-up, respectively. However, we could not confirm the precise reason that these factors are negative predictors because such information is not available from the claims data; this is another limitation of this study. Furthermore, although the patients with these negative predictors were more likely to die of liver cancer, official cause of death is not recorded in the claims data, and such causal links could not be confirmed in this study.

Finally, this was an observational cohort study, and its validity may be jeopardised by confounding variables, such as cancer stage, body mass index, smoking and alcohol consumption, that were not included in the dataset and thus were not measured. We therefore suggest that randomised clinical trials be conducted before the conclusions of the present study are applied to clinical practice in Taiwan and elsewhere.

## CONCLUSIONS

The 5-year survival probability of patients with liver cancer may improve if they receive abdominal US three times or more during the 2 years preceding their liver cancer diagnosis. Additional studies should prospectively investigate whether abdominal US can be used to identify early-stage liver cancer and examine whether further treatment with regular follow-up improves survival.

**Contributors** JKC, LCW and YHK designed, conducted and drafted the manuscript. JKC analysed the data. All authors contributed to the manuscript, revised drafts critically for important intellectual content and read and approved the final manuscript.

**Funding** JKC received research grants from Dalin Tzu Chi Hospital, Buddhist Tzu Chi Medical Foundation (DTCRD105 (2)E22). The funding source had no involvement in the study design; collection, analysis and interpretation of data; writing of the report; or decision to submit the article for publication.

**Disclaimer** This study is based in part on data from the National Health Insurance Research Database, provided by the Bureau of National Health Insurance of the Department of Health and managed by the National Health Research Institutes. The interpretations and conclusions contained herein do not represent those of the Bureau of National Health Insurance, Department of Health or National Health Research Institutes.

**Competing interests** None declared.

**Ethics approval** The study protocol was reviewed and approved by the Research Ethics Committee of Dalin Tzu Chi Hospital, Buddhist Tzu Chi Medical Foundation, Taiwan (approval no B10304018).

**Provenance and peer review** Not commissioned; externally peer reviewed.

**Data sharing statement** Extra data can be accessed via the Dryad data repository at http://datadryad.org/ with the doi:10.5061/dryad.r1m19.

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
