## [Reviewer comments · BMJ Open]

ARTICLE DETAILS

TITLE (PROVISIONAL)	Effect of Ultrasonography Surveillance in Patients with Liver Cancer: A Population-Based Longitudinal Study
AUTHORS	Chiang, Jui-Kun; Lin, Chih-Wen; Kao, Yee-Hsin

VERSION 1 - REVIEW

REVIEWER	Jing-Houng Wang Kaohsiung Chang Gung Memorial Hospital, Kaohsiung City, Taiwan
REVIEW RETURNED	25-Jan-2017

GENERAL COMMENTS	General comments This study by Chiang et al. analyzed effect of ultrasonography (US) frequency 2 years prior to diagnosis on survival for patients with liver cancer. In this population-based study with 4621 patients with liver cancer from health insurance research data in Taiwan, those patients with liver cancer receiving ≥ 3 times US (M3) within 2 years prior diagnosis had higher survival time (1.4 years vs 0.51 years) and rate (14.4% vs 7.7%) compared to those with less than 3 US exam (L3). More patients in the M3 group received treatments for early stage hepatoma. More patients with chronic hepatitis received more than 3 times of US exam 2 years prior diagnosis of liver cancer. The authors concluded higher survival probability for patients in M3 group. Some issues should be clarified in this study. Major comments 1. Is there a dose-response effect between US frequency and survival?2. Compared to M3 group, low survival rate and time for patients in L3 group might be related to acute disease not related to liver cancer. The cause of patient death might be important in this study to make the conclusion more convincing for the readers.3. In the discussion section, the authors suggested that more than three US surveillance within 2 years for general population in high endemic area of liver cancer. This suggestion might be too early for lacking cost-effective analysis in this study.4. Similar study (Wu CY, et al. GUT 2016) also concluded that shorter US screening reduced overall mortality of hepatoma in a nationwide cohort study from the same area. Is the database in this study the same with the published study?
---

REVIEWER	Amedeo Lonardo AOU Modena
REVIEW RETURNED	19-Apr-2017

GENERAL COMMENTS	GENERAL COMMENT This is an ingenious epidemiological study which not only
--

demonstrates that receiving more intensive US surveillance will be associated with improved surveillance but also that this occurs owing to these patients receiving earlier diagnosis and more appropriate treatment schedules for HCC. I would suggest widening the scope of the discussion and adding further references.

**SPECIFIC COMMENT
MAJOR**

I found the definitions "L3" and "M3" to be not particularly reader-friendly. Would suggest "<3" and ">3" throughout the manuscript, tables included.

This article shows that the indications in L3 and M3 groups are different (Table 2). Could Authors go further in discussing whether, in clinical practice, all individuals with chronic hepatitis (due to either viral or non-viral etiology); pancreas disease (is this due to alcohol abuse ? Or does this occur owing to T2D ?) and KUB should undergo more intensive US surveillance ?

This article adds to the list of previous articles suggesting that certain populations at a high-risk for developing HCC will benefit from more intensive US surveillance. Please, cite the following articles: Santi V, J Hepatol. 2010;53:291-7. Santi V, J Hepatol. 2012;56:397-405. Giannini EG, World J Gastroenterol. 2013;19:8808-21. Del Poggio P, Clin Gastroenterol Hepatol. 2014;12:1927-33.e2. Cucchetti A, J Hepatol. 2014;61:333-41.

Table 5 . The finding that male gender and T2D are independently associated with M3 should also be further developed. For example: are any data available regarding the duration of diabetes ? Moreover, how was T2D treated in these patients ? Data suggest that insulin might enhance and certain oral glucose-lowering medications and statins possibly decrease the risk of HCC (Lonardo A, Expert Rev Gastroenterol Hepatol. 2015;9:629-50. Lonardo A, J Gastroenterol Hepatol. 2012;27:1654-64). Finally, Authors may be willing to discuss that male gender and T2D are potential clues to the diagnosis of NAFLD (Non-alcoholic Fatty Liver Disease Study Group., Dig Liver Dis. 2015;47:997-1006). Recent contributions on NASH-HCC should be cited (Rinaldi L, World J Gastroenterol. 2017;23:1458-1468. Piscaglia F, Hepatology. 2016;63:827-38.).

The limitations of the study need to be further argued. For example this is not a randomized trial but an observational study which is, thus, exposed to several bias. Moreover, data on duration and drug treatment of T2D might be incomplete. Having said that, how applicable are its conclusions to clinical practice in Taiwan and elsewhere?

MINOR

Table 2: What does KUB stand for ? Please specify at the foot of the table. Conversely, are the following explanations necessary ? CKD, chronic kidney disease; CVA, cerebral vascular accident; CCI, Charlson comorbidity index; DU, duodenal ulcer; GU, gastric ulcer; EV; esophageal varices; US, ultrasonography surveillances

VERSION 1 – AUTHOR RESPONSE

General comments

This study by Chiang et al. analyzed effect of ultrasonography (US) frequency 2 years prior to diagnosis on survival for patients with liver cancer. In this population-based study with 4621 patients with liver cancer from health insurance research data in Taiwan, those patients with liver cancer receiving ≥ 3 times US (M3) within 2 years prior diagnosis had higher survival time (1.4 years vs 0.51 years) and rate (14.4% vs 7.7%) compared to those with less than 3 US exam (L3). More patients in the M3 group received treatments for early stage hepatoma. More patients with chronic hepatitis received more than 3 times of US exam 2 years prior diagnosis of liver cancer. The authors concluded higher survival probability for patients in M3 group. Some issues should be clarified in this study.

Major comments

1. Is there a dose-response effect between US frequency and survival?

Response: Thank you for your advice! Yes, as shown in Figure 2 of the manuscript, there was a linear dose-response relationship between US frequency and logit of 5-year survival probability. We add this point to the discussion section (page 14, line 1 to line 4). A similar result was also reported in previous study.²³

New reference:

23. Wu CY, Hsu YC, Ho HJ, et al. Association between ultrasonography screening and mortality in patients with hepatocellular carcinoma: a nationwide cohort study. *Gut* 2016; 65: 693-701.

2. Compared to M3 group, low survival rate and time for patients in L3 group might be related to acute disease not related to liver cancer. The cause of patient death might be important in this study to make the conclusion more convincing for the readers.

Response: Thank you for your advice! Although such patients more likely died of liver cancer, the cause of death was not recorded in claim data, and thus it was one of the limitations in this study. Hence, we add this point in the Discussion section (page 18, line 9 to line 11).

3. In the discussion section, the authors suggested that population in high endemic area of liver cancer. This suggestion might be too early for lacking cost-effective analysis in this study.

Response: Thank you for your advice! As you suggested, we modify the original sentence. Further cost-effectiveness analyses are needed to justify the recommendation of ultrasonography surveillance three times or more within 2 years to high risk people or general population in the area of high incidence of liver cancer (page 14, line 4 to line 6).

4. Similar study (Wu CY, et al. GUT 2016) also concluded that shorter US screening reduced overall mortality of hepatoma in a nationwide cohort study from the same area. Is the database in this study the same with the published study?

Response: Thank you for your advice! Both studies used the data from the National Health Insurance Research Database (NHIRD), which was made available by the National Health Insurance Administration, Ministry of Health and Welfare.

However, the data used in this study was different from that of the previous published study in the inclusion criteria and sub-databases retrieved from the NHIRD. In the previous study, they identified all newly diagnosed HCC patients admitted with a primary diagnosis of HCC or enrolled in the

Registry for Catastrophic Illness Patient Database between 1 January 2002 and 31 December 2007. In this study, we identified first-time diagnosed liver cancer patients, aged ≥ 18 years, with catastrophic illness designations between January 1, 1997, and December 31, 2010 in the Longitudinal Health Insurance Database 2000 (LHID2000) and then followed up these patients until December 2011.

Reviewer: 2

Amedeo Lonardo

AOU Modena

Please state any competing interests or state 'None declared': 'None declared'

Please leave your comments for the authors below

bmjopen-2017-015936

GENERAL COMMENT

This is an ingenious epidemiological study which not only demonstrates that receiving more intensive US surveillance will be associated with improved surveillance but also that this occurs owing to these patients receiving earlier diagnosis and more appropriate treatment schedules for HCC. I would suggest widening the scope of the discussion and adding further references.

Response: Thank you for your advice! We add sentences and new references in the Discussion section.

page 14, line 11 to line 13:

Some previous articles also reported that certain populations at a high-risk for developing HCC will benefit from more intensive US surveillance.²⁶⁻³⁰

page 16, line 5 to line 6:

Data suggest that insulin might enhance and certain oral glucose-lowering medications and statins possibly decrease the risk of HCC. ^{39 40}

page 16, line 15 to page 17, line 4:

Diabetes, hypertension, obesity, metabolic syndrome, mixed hyperlipidemia and hypocholesterolemia due to familial hypobetalipoproteinemia were the non-alcoholic fatty liver disease risk.⁴³ Non-alcoholic fatty liver disease has an etiology role in cryptogenic cirrhosis that is associated with a poor prognosis and high risk of cardiovascular disease and early HCC development.^{44 45} In this study, the non-alcoholic fatty liver was included in group of liver-related diseases including chronic hepatitis, cirrhosis, and other liver disorders. The precise classification for non-alcoholic fatty liver was also one of the limitations of this study originated from claim data.

New references:

26. Santi V, Trevisani F, Gramenzi A et al. Semiannual surveillance is superior to annual surveillance for the detection of early hepatocellular carcinoma and patient survival. *J Hepatol* 2010; 53: 291-297.

27. Santi V, Buccione D, Di Micoli A et al. The changing scenario of hepatocellular carcinoma over the last two decades in Italy. *J Hepatol* 2012; 56: 397-405.

28. Giannini EG, Cucchetti A, Erroi V et al. Surveillance for early diagnosis of hepatocellular carcinoma: how best to do it? *World J Gastroenterol* 2013; 19: 8808-8821.

29. Del Poggio P, Olmi S, Ciccarese F et al. Factors that affect efficacy of ultrasound surveillance for

early stage hepatocellular carcinoma in patients with cirrhosis. Clin Gastroenterol Hepatol 2014; 12: 1927-1933 e1922.

30. Cucchetti A, Trevisani F, Pecorelli A et al. Estimation of lead-time bias and its impact on the outcome of surveillance for the early diagnosis of hepatocellular carcinoma. J Hepatol 2014; 61: 333-341.

39. Lonardo A, Ballestri S, Targher G, Loria P. Diagnosis and management of cardiovascular risk in nonalcoholic fatty liver disease. Expert Rev Gastroenterol Hepatol 2015; 9: 629-650.

40. Lonardo A, Loria P. Potential for statins in the chemoprevention and management of hepatocellular carcinoma. J Gastroenterol Hepatol 2012; 27: 1654-1664.

43. Non-alcoholic Fatty Liver Disease Study G, Lonardo A, Bellentani S et al. Epidemiological modifiers of non-alcoholic fatty liver disease: Focus on high-risk groups. Dig Liver Dis 2015; 47: 997-1006.

44. Rinaldi L, Nascimbeni F, Giordano M et al. Clinical features and natural history of cryptogenic cirrhosis compared to hepatitis C virus-related cirrhosis. World J Gastroenterol 2017; 23: 1458-1468.

45. Piscaglia F, Svegliati-Baroni G, Barchetti A et al. Clinical patterns of hepatocellular carcinoma in nonalcoholic fatty liver disease: A multicenter prospective study. Hepatology 2016; 63: 827-838.

SPECIFIC COMMENT

MAJOR

I found the definitions "L3" and "M3" to be not particularly reader-friendly. Would suggest "<3" and ">3" throughout the manuscript, tables included.

Response: Thank you for your advice! We change the "L3" and "M3" to "<3" and "≥3" respectively.

This article shows that the indications in L3 and M3 groups are different (Table 2). Could Authors go further in discussing whether, in clinical practice, all individuals with chronic hepatitis (due to either viral or non-viral etiology); pancreas disease (is this due to alcohol abuse? Or does this occur owing to T2D?) and KUB should undergo more intensive US surveillance ?

Response: Thank you for your advice!

In this study, we classified the diseases that are the indications for ultrasonography as (1) liver-related diseases including chronic hepatitis, cirrhosis, and other liver disorders (ICD-9 codes: 571, 571.x, 572, 572.0, 573, and 573.x), (5) kidney–urinary–bladder (KUB) diseases (ICD-9 codes: 591, 592, 592.0, 592.1, 592.9, 593, and 593.2); (6) pancreatic diseases (ICD-9 code: 577.x). Patients with above indications could receive US surveillance due to US surveillance was easily available in Taiwan. The underlying cause of pancreatic disease included alcohol abuse, biliary disease, hyperlipidemia, and other risk factors, etiologies or idiopathic disease.⁴⁶ It is another limitation in this study, the baseline data were not recorded in the claim data for investigate. (page 17, line 4 to page 10)

New reference:

46. Chen CH, Dai CY, Hou NJ, et al. Etiology, severity and recurrence of acute pancreatitis in southern taiwan. J Formos Med Assoc 2006; 105: 550-555.

This article adds to the list of previous articles suggesting that certain populations at a high-risk for developing HCC will benefit from more intensive US surveillance. Please, cite the following articles:

Santi V, J Hepatol. 2010;53:291-7. Santi V, J Hepatol. 2012;56:397-405. Giannini EG, World J Gastroenterol. 2013;19:8808-21. Del Poggio P, Clin Gastroenterol Hepatol. 2014;12:1927-33.e2. Cucchetti A, J Hepatol. 2014;61:333-41.

Response: Thank you for your advice! We add the following point and new references in the Discussion section (page 14, line 11 to line 13).

Some previous articles also reported that certain populations at a high-risk for developing HCC will benefit from more intensive US surveillance.²⁶⁻³⁰

New references:

26. Santi V, Trevisani F, Gramenzi A, et al. Semiannual surveillance is superior to annual surveillance for the detection of early hepatocellular carcinoma and patient survival. J Hepatol 2010; 53: 291-297.
27. Santi V, Buccione D, Di Micoli A, et al. The changing scenario of hepatocellular carcinoma over the last two decades in Italy. J Hepatol 2012; 56: 397-405.
28. Giannini EG, Cucchetti A, Erroi V, et al. Surveillance for early diagnosis of hepatocellular carcinoma: how best to do it? World J Gastroenterol 2013; 19: 8808-8821.
29. Del Poggio P, Olmi S, Ciccarese F, et al. Factors that affect efficacy of ultrasound surveillance for early stage hepatocellular carcinoma in patients with cirrhosis. Clin Gastroenterol Hepatol 2014; 12: 1927-1933 e1922.
30. Cucchetti A, Trevisani F, Pecorelli A, et al. Estimation of lead-time bias and its impact on the outcome of surveillance for the early diagnosis of hepatocellular carcinoma. J Hepatol 2014; 61: 333-341.

Table 5. The finding that male gender and T2D are independently associated with M3 should also be further developed. For example: are any data available regarding the duration of diabetes? Moreover, how was T2D treated in these patients? Data suggest that insulin might enhance and certain oral glucose-lowering medications and statins possibly decrease the risk of HCC (Lonardo A, Expert Rev Gastroenterol Hepatol. 2015; 9:629-50. Lonardo A, J Gastroenterol Hepatol. 2012; 27:1654-64). Finally, Authors may be willing to discuss that male gender and T2D are potential clues to the diagnosis of NAFLD (Non-alcoholic Fatty Liver Disease Study Group., Dig Liver Dis. 2015;47:997-1006). Recent contributions on NASH-HCC should be cited (Rinaldi L, World J Gastroenterol. 2017; 23:1458-1468. Piscaglia F, Hepatology. 2016; 63:827-38.).

Response: Thank you for your advice! We add the references after the sentences of in the Discussion section.

page 16, line 5 to line 6:

Data suggest that insulin might enhance and certain oral glucose-lowering medications and statins possibly decrease the risk of HCC. 39 40

page 16, line 15 to page 17, line 4:

Diabetes, hypertension, obesity, metabolic syndrome, mixed hyperlipidemia and hypocholesterolemia due to familial hypobetalipoproteinemia were the non-alcoholic fatty liver disease risk.⁴³ Non-alcoholic fatty liver disease has an etiology role in cryptogenic cirrhosis that is associated with a poor prognosis and high risk of cardiovascular disease and early HCC development.^{44 45} In this study, the non-alcoholic fatty liver was included in group of liver-related diseases including chronic hepatitis, cirrhosis, and other liver disorders. The precise classification for non-alcoholic fatty liver was also one of the limitations of this study originated from claim data.

New references:

39. Lonardo A, Ballestri S, Targher G, et al . Diagnosis and management of cardiovascular risk in nonalcoholic fatty liver disease. Expert Rev Gastroenterol Hepatol 2015; 9: 629-650.

40. Lonardo A, Loria P. Potential for statins in the chemoprevention and management of hepatocellular carcinoma. *J Gastroenterol Hepatol* 2012; 27: 1654-1664.
43. Non-alcoholic Fatty Liver Disease Study G, Lonardo A, Bellentani S, et al. Epidemiological modifiers of non-alcoholic fatty liver disease: Focus on high-risk groups. *Dig Liver Dis* 2015; 47: 997-1006.
44. Rinaldi L, Nascimbeni F, Giordano M, et al. Clinical features and natural history of cryptogenic cirrhosis compared to hepatitis C virus-related cirrhosis. *World J Gastroenterol* 2017; 23: 1458-1468.
45. Piscaglia F, Svegliati-Baroni G, Barchetti A, et al. Clinical patterns of hepatocellular carcinoma in nonalcoholic fatty liver disease: A multicenter prospective study. *Hepatology* 2016; 63: 827-838.

The limitations of the study need to be further argued. For example, this is not a randomized trial but an observational study which is, thus, exposed to several bias. Moreover, data on duration and drug treatment of T2D might be incomplete. Having said that, how applicable are its conclusions to clinical practice in Taiwan and elsewhere?

Response: Thank you for your advice! We add the following point to the Discussion section as a study limitation (pages 18, line 11 to line 15):

This was an observational cohort study, and thus its validity might be jeopardized by confounding biases of unmeasured or unknown covariates. Hence, if it is feasible, randomized clinical trials should be conducted before its conclusions are applied to clinical practice in Taiwan and elsewhere.

MINOR

Table 2: What does KUB stand for? Please specify at the foot of the table. Conversely, are the following explanations necessary? CKD, chronic kidney disease; CVA, cerebral vascular accident; CCI, Charlson comorbidity index; DU, duodenal ulcer; GU, gastric ulcer; EV; esophageal varices; US, ultrasonography surveillances

Response: Thank you for your advice! As you suggested, we add the abbreviation of KUB, "kidney-urinary-bladder," in the bottom of Table 2 and deleted the other well-known abbreviations.

VERSION 2 – REVIEW

REVIEWER	Amedeo Lonardo Nuovo Ospedale Civile di Baggiovara, Modena, Italy
REVIEW RETURNED	10-May-2017

GENERAL COMMENTS	Authors satisfactorily addressed all topics raised by this Reviewer. However, I found the following sente unclear "Diabetes, hypertension, obesity, metabolic syndrome, mixed hyperlipidemia and hypocholesterolemia due to familial hypobetalipoproteinemia were the non-alcoholic fatty liver disease risk". Would suggest rephrasing as follows: "The chances of developing NAFLD are enhanced owing to T2D, arterial hypertension, obesity, metabolic syndrome, mixed hyperlipidemia and familial hypobetalipoproteinemia, which are the chief metabolic modifiers of NAFLD risk".
--

VERSION 2 – AUTHOR RESPONSE

Reviewer: 2

Amedeo Lonardo

Nuovo Ospedale Civile di Baggiovara, Modena, Italy

Please state any competing interests or state 'None declared': None declared

Please leave your comments for the authors below

Authors satisfactorily addressed all topics raised by this Reviewer. However, I found the following sentence unclear "Diabetes, hypertension, obesity, metabolic syndrome, mixed hyperlipidemia and hypocholesterolemia due to familial hypobetalipoproteinemia were the non-alcoholic fatty liver disease risk". Would suggest rephrasing as follows: "The chances of developing NAFLD are enhanced owing to T2D, arterial hypertension, obesity, metabolic syndrome, mixed hyperlipidemia and familial hypobetalipoproteinemia, which are the chief metabolic modifiers of NAFLD risk".

Response: Thank you for your advice!

We revised the sentence as your suggestion.

Page 17, lines 9 to lines 12

The chances of developing NAFLD are enhanced by type 2 diabetes, arterial hypertension, obesity, metabolic syndrome, mixed hyperlipidemia, and familial hypobetalipoproteinemia, all of which are the chief metabolic modifiers of NAFLD risk.43- 45